# Evaluating Time Series Foundation Models for Electricity Price Forecasting: Contamination Risk, Distributional Shifts, and Covariate Dependence

**Zhenghua Pan** [1]   **Ahmed Aziz Ezzat** [1]

## Abstract

Time series foundation models (TSFMs) have shown strong zero-shot forecasting performance, but their generalization in covariate-driven, non-stationary settings is underexplored. Electricity price forecasting (EPF) presents a challenging testbed due to complex temporal dependencies, distributional shifts, and strong reliance on structural and contextual information. We propose a two-dataset-benchmarking framework for EPF to mitigate contamination risk and enable fair evaluation of TSFMs. We examine key aspects of EPF including point and probabilistic forecasting performance, tail behavior, price spikes, and comparisons against domain-specific methods. We find that TSFMs are highly competitive and often outperform general-purpose baselines. Yet, their performance depends critically on covariate support, and they do not consistently surpass domain-specific methods tailored to EPF. Interestingly, simple ensembles of TSFMs and domain-specific methods appear to have significant potential, suggesting that the two approaches capture complementary predictive information.

## 1. Introduction

Short-term electricity price forecasting (EPF) is a cornerstone of modern power system operations. Yet, it remains highly challenging due to strong nonstationarity, multi-scale variability (e.g., seasonality, spikes, and extreme events), as well as dependence on complex structural and contextual system dynamics (Weron, 2014; Ezzat et al., 2026).

The literature on EPF is extensive (Weron, 2014; Maciejowska et al., 2022), spanning classical autoregressive and time series approaches (Conejo et al., 2005; Zareipour et al., 2006; Ziel & Weron, 2018), tree- and ensemble-based methods (Wang et al., 2022; Mei et al., 2014), and more recently, deep learning models (Xie et al., 2018; Lago et al., 2018; Zhang & Wu, 2023).

Recent advances in Time Series Foundation Models (TSFMs) offer new opportunities for improved short-term EPF. Several recent studies have benchmarked TSFMs across a variety of forecasting domains (Das et al., 2023b; Garza et al., 2023; Zhang et al., 2024; Woo et al., 2024; Li et al., 2025; Qiu et al., 2024; Aksu et al., 2024; Ansari et al., 2024; Shchur et al., 2025; Meyer et al., 2025)—See Table 2 (Appendix A.1) for a comparative list and where our study stands. Yet, their performance in EPF remains largely underexplored. Benchmarking TSFMs in EPF raises four key aspects: (*i*) contamination risk, since many TSFMs are pretrained on energy-related data; (*ii*) nonstationarity, particularly in the form of distributional shifts and price spikes that can undermine model generalization; (*iii*) tail behavior, which is critical for forecast-informed decision-making in power systems; and (*iv*) strong dependence on exogenous covariates and contextual information, since historical prices alone are often insufficient for accurate EPF. Recently, Hornek et al. (2025) benchmarked TSFMs for EPF across several electricity markets. Our study builds on, and departs from, this line of research in several key respects. First, we evaluate recent TSFMs and examine the role of exogenous covariates, comparing univariate and covariate-supported variants. Second, we benchmark TSFMs not only against statistical baselines, but also against deep learning and domain-specific methods tailored to EPF. Third, we extend the evaluation beyond average forecast accuracy to include probabilistic performance, tail behavior, and distributional shifts. Finally, we introduce a two-dataset evaluation protocol to explicitly consider contamination risk.

The challenges of EPF motivate the need for a dedicated benchmarking framework. A central question is *how pretrained TSFMs compare not only against general-purpose baselines, but also against domain-specific methods carefully designed around the unique characteristics of electricity pricing signals?* In response, we propose a two-dataset benchmarking framework that accounts for contamination risk and enables fair evaluation of TSFMs. We

---

[1]Department of Industrial & Systems Engineering, Rutgers, The State University of New Jersey, USA. Correspondence to: Ahmed Aziz Ezzat <aziz.ezzat@rutgers.edu>.

*ICML 2026 Foundation Models for Structured Data Workshop, 43$^{rd}$ International Conference on Machine Learning (ICML),* Seoul, South Korea. Copyright 2026 by the author(s).

further examine key aspects of EPF performance, including point and probabilistic forecasting, tail behavior, and robustness to price spikes. Across these aspects, we benchmark TSFMs against a wide range of statistical, deep learning, and domain-specific methods, allowing us to assess whether large-scale pretraining can complement or replace task-specific domain design.

## 2. Benchmarking Framework

### 2.1. The forecasting setup: Day-ahead EPF

We formalize the EPF task under a day-ahead electricity market setup. Let $t$ denote the forecast origin and $t + \delta_t$ denote the first hour of the forecast horizon, where $\delta_t$ is the time gap between forecast issuance (pre-market closure) and the first hour of the forecast horizon. The forecasting objective is to estimate the following predictive distribution:

$$\mathbb{P}\left(Y_{t+\delta_t:t+\delta_t+H-1} | \mathcal{F}_t\right), \tag{1}$$

where $Y_j$ is the pricing signal at time $j$, and $\mathcal{F}_j$ denotes all information available at time $j$, including realized and cleared prices, as well as historical and forecasted electric load, fuel prices, generation, weather, and calendar features, etc.. Here, $L$ is the size of historical data and $H = 24$ is the forecast horizon. The covariate-free setting is recovered when only past and cleared prices are used in the set $\mathcal{F}_j$.

### 2.2. Model Selection

We benchmark four TSFMs: *Chronos 2* (Ansari et al., 2025), *TimesFM 2.5* (Das et al., 2023b), *TabPFN-TS* (Hoo et al., 2025), and *TOTO 1.0* (Cohen et al., 2025). For each TSFM, we evaluate both covariate-free and covariate-supported variants to assess the importance of exogenous information in EPF. Hereinafter, we denote covariate-supported variants with "w." (e.g., *Chronos 2 w.*), while covariate-free variants do not have this suffix. Model selection was guided by three criteria: (*i*) covariates support ; (*ii*) probabilistic forecast output; and (*iii*) accessibility and reproducibility. Additionally, we focus on zero-shot rather than few-shot forecasting to ensure consistent evaluation across selected TSFMs.

We compare TSFMs against three classes of baseline methods. First, we consider classical time series models, including: *Seasonal Naïve* (Hyndman & Athanasopoulos, 2018), *Auto-ARIMA*, *Auto-ETS*, and *Auto-Theta* (Garza et al., 2022). Second, we consider task-trained deep learning approaches, including *DeepAR* (Salinas et al., 2020), *TFT* (Lim et al., 2021), *TiDE* (Das et al., 2023a), *TSMixer* (Chen et al., 2023), *DLinear*, and *NLinear* (Zeng et al., 2023). Third, we benchmark against three domain-specific EPF methods, namely *LEAR* (Uniejewski et al., 2016), *DNN* (Lago et al., 2018), and *CING-LEAR* (Wang et al., 2026), which are explicitly designed to leverage electricity market information.

### 2.3. Two-Dataset Benchmarking Framework

Data contamination is a major challenge in evaluating TSFMs (Aksu et al., 2024; Meyer et al., 2025; Shchur et al., 2025) because pretraining corpora are large, heterogeneous, and often not fully or clearly disclosed. In addition to direct exposure, TSFMs are also prone to temporal contamination (Meyer et al., 2026). As a result, it is difficult to determine whether benchmark datasets overlap with pretraining data, potentially leading to inflated performance estimates. Effective mitigation techniques remain limited in large-scale benchmarking due to the unique pretraining corpus curated for each TSFM. Unlike live-evaluation platforms, such as TS-ARENA (Meyer et al., 2025), we evaluate TSFMs on two complementary datasets: an established, widely used benchmark dataset; and another, recently curated dataset selected to minimize overlap with TSFM pretraining corpora.

Specifically, we evaluate all models on (*i*) GEFCom2014-P (Hong et al., 2016)—the probabilistic EPF track of the 2014 Global Energy Forecasting Competition, and (*ii*) GridStatus2025, a new benchmark dataset that we have curated from the GridStatus API (GridStatus, 2026). GEFCom2014-P serves as a standardized and widely used EPF benchmark, while GridStatus2025 was selected to reduce potential temporal overlap. Based on publicly disclosed information, the real-world pretraining data for *Chronos 2* and *TimesFM 2.5* predates the evaluation period for the GridStatus2025 dataset, while *TabPFN-TS* relies on synthetic pretraining data. Although TOTO 1.0 incorporates proprietary data whose temporal coverage is not publicly disclosed, its technical documentation predates the evaluation period of the GridStatus2025 dataset (Cohen et al., 2024). Hence, this two-dataset benchmarking design provides a reasonably reliable assessment of model generalization while mitigating potential contamination effects.

### 2.4. Evaluation Protocols

For GEFCom2014-P, we follow the rolling evaluation protocol proposed by the competition organizers. The competition consists of multiple forecasting tasks, where each task corresponds to one rolling forecast origin: contestants are given data available up to forecast origin for training and then asked to predict the 24 hourly electricity prices for the next day. We evaluate TSFMs on nine quantiles $\{0.1, 0.2, \ldots, 0.9\}$ with performance averaged uniformly across tasks. For GridStatus2025, we perform day-ahead hourly forecasting using three years of training data (2022–2024) and one year of testing data (2025). Covariates include load, solar, gas and fuel prices. All models are evaluated on a rolling basis using Mean Absolute Error (MAE), Root Mean Square Error (RMSE), and average quantile loss (aQL) over nine quantiles (See metric and implementation details in A.2 and A.3, respectively).

Beyond average forecast accuracy, we examine several dimensions that are critical to EPF. First, we evaluate probabilistic forecasting performance through quantile-based scores. Second, we analyze tail behavior to examine model performance under extreme price conditions. Finally, we evaluate robustness under nonstationarity through dedicated price spike analyses, where abrupt market or system changes induce significant distributional shifts. We use GEFCom2014-P as a standardized benchmark to compare zero-shot TSFMs against official competition submissions. However, since complete code and prediction outputs of participants are not publicly available, controlled follow-up analyses are not possible. Consequently, detailed diagnostic analyses are conducted on GridStatus2025, where all models are implemented within a unified evaluation pipeline.

## 3. Results and Discussions

### 3.1. Average Performance and Covariate Support

On the GEFCom2014-P price track (Table 4, Appendix A.4), TSFMs achieve competitive zero-shot performance relative to many competition submissions, without task-specific training nor feature engineering. However, their performance remains below the strongest specialized entries: none of the evaluated TSFMs reaches the top-5, and the best-performing TSFM, *TOTO 1.0*, ranks eighth overall. These results suggest that, although large-scale pretraining provides strong general-purpose forecasting capability, domain-specific methods remain advantageous for EPF, where accurate prediction relies on exogenous information, and EPF-orientated feature and architecture design.

We further evaluate models on the more recent GridStatus2025 dataset (Table 1). Here, TSFMs generally outperform statistical and deep learning baselines, with covariate-supported *Chronos 2 w.* and *TabPFN-TS w.* consistently ranking among the strongest TSFMs across all metrics. In contrast, covariate-free variants perform substantially worse. Taken together with Diebold-Mariano (DM) tests (Diebold & Mariano, 1995) in Table 7 (Appendix A.5), these results reinstate the importance of exogenous information in EPF. At the same time, domain-specific methods remain highly competitive. Specifically, *CING-LEAR* achieves the second-best individual performance on GridStatus2025, outperforming most TSFMs and general-purpose forecasting models. Interestingly, a simple ensemble, constructed by averaging the forecasts of the best-performing TSFM (*Chronos 2 w.*) and domain-specific model (*CING-LEAR*) achieves the strongest overall performance, yielding considerable gains over both constituent models. This suggests that pre-trained TSFMs and domain-aware forecasting methods capture diverse and complementary predictive information that can be effectively combined for improved predictive skill.

We observe that *TOTO* exhibits strong performance on GEFCom2014-P but degrades substantially on GridStatus2025. While several factors may contribute to this discrepancy, including differences in market conditions and data characteristics, it highlights the importance of contamination-conscious evaluation.

*Table 1.* Average (point) forecast performance on the GridStatus2025 dataset. Bold-faced and underlined values denote best and second best performance, respectively. Rank denotes the average ranking across MAE, RMSE, and aQL. Green color indicates lowest average rank, red color indicates highest average rank.

| Category | Model | MAE | RMSE | aQL | Rank |
|---|---|---|---|---|---|
| STATISTICAL | SEASONAL NAIVE | 5.791 | 8.491 | – | – |
| | AUTO-ARIMA | 4.638 | 6.540 | 1.960 | 6.00 |
| | AUTO-THETA | 7.912 | 10.844 | 3.836 | 16.00 |
| | AUTO-ETS | 9.506 | 13.270 | 4.013 | 17.33 |
| DEEP LEARNING | TiDE | 5.683 | 7.441 | 2.357 | 11.67 |
| | TSMIXER | 6.624 | 8.516 | 2.546 | 13.00 |
| | DEEPAR | 7.541 | 10.056 | 3.090 | 14.00 |
| | TFT | 7.837 | 10.644 | 3.100 | 15.00 |
| | DLINEAR | 9.646 | 12.503 | 4.072 | 17.67 |
| | NLINEAR | 10.153 | 13.381 | 4.119 | 19.00 |
| TSFMs | CHRONOS 2 W. | 4.105 | 5.719 | 1.631 | 2.00 |
| | TABPFN-TS W. | 4.521 | 6.502 | 1.787 | 4.00 |
| | CHRONOS 2 | 4.698 | 6.811 | 1.866 | 5.67 |
| | TIMESFM 2.5 W. | 4.982 | 6.931 | 1.947 | 8.00 |
| | TIMESFM 2.5 | 4.972 | 7.235 | 1.968 | 9.00 |
| | TABPFN-TS | 5.320 | 7.648 | 2.083 | 11.33 |
| | TOTO 1.0 | 11.542 | 16.208 | 5.084 | 20.00 |
| | TOTO 1.0 W. | 11.795 | 16.539 | 5.160 | 21.00 |
| DOMAIN-AWARE | CING-LEAR | 4.202 | 5.843 | 1.664 | 3.00 |
| | DNN | 4.942 | 6.822 | 1.943 | 6.67 |
| | LEAR | 5.019 | 7.092 | 1.987 | 9.67 |
| ENSEMBLE | CHRONOS 2 W. + CING-LEAR | **3.922** | **5.470** | **1.541** | **1.00** |

### 3.2. Probabilistic and Tail Performance

In EPF, probabilistic and tail performance are central to risk-aware decision-making since forecast errors often have asymmetric financial risks (Pinson, 2023). Here, we focus on GridStatus2025 and evaluate whether models can provide sharp and calibrated probabilistic forecasts.

Probabilistic performance is assessed using aQL defined in Equation 4 (Appendix A.2). As shown in Table 1, TSFMs achieve strong quantile-based performance compared with statistical and deep learning baselines, suggesting that pre-trained models better capture distributional information in EPF. In particular, *Chronos 2 w.* and *TabPFN-TS w.* achieve the best aQL among TSFMs, again highlighting the importance of covariate support. *CING-LEAR* achieves the second-lowest overall aQL among individual models, indicating that domain and contextual information are key for accurate probabilistic forecasting. The simple ensemble clearly outperforms all methods, reinstating that diverse forecast combinations can enhance distributional accuracy.

We further analyze model behavior across tail quantiles to assess robustness under extreme price conditions. The results shown in Table 5 (Appendix A.4) demonstrate that median performance ($Q(0.50)$) alone does not fully characterize forecasting behavior in the tails. Several models that perform competitively around $Q(0.50)$ exhibit substantial

degradation in tail regions (especially lower tails). This is especially evident for statistical forecasting models, whose lower-tail losses (e.g., $Q(0.01)$, $Q(0.025)$, $Q(0.05)$) increase significantly relative to TSFMs and domain-specific methods. These findings highlight that average predictive accuracy can obscure important weaknesses in modeling rare low-price events, including negative electricity prices.

We also observe a pronounced asymmetry between forecast performance in two tails. Statistical methods generally perform better in upper-tail while struggling in the lower tail, suggesting difficulty in capturing abrupt downward price movements. TSFMs produce more balanced tail forecasts across both extremes, indicating improved distributional robustness. Domain-specific methods, especially *CING-LEAR*, are highly competitive across all tail quantiles. Notably, the ensemble model achieves the best performance across almost all quantiles.

### 3.3. Performance Under Distributional Shifts

Price spikes are localized distributional shifts that are difficult to predict, and are often associated with substantial financial risks. Here, we evaluate forecast robustness to price spikes on GridStatus2025. Following prior studies (Christensen et al., 2012; Mount et al., 2006), we define spikes as observations that fall below the 5th percentile or above the 95th percentile of the test set price distribution, capturing abnormally low and high price regimes.

Figure 1 compares point forecasts, along with 80% prediction intervals, for the best-performing methods from each model family during a representative spike (top panel) and non-spike period (bottom panel). Overall, *Chronos 2 w.* and *CING-LEAR* appear to better track price dynamics during most price spike events (see, e.g., period from Jan 19 to 21), and so does their ensemble. During non-spike periods, both models maintain reasonably accurate forecasts. TiDE appears to persistently over- and under-estimate pricing signals in both spike and non-spike regimes, despite its relatively moderate forecast degradation in terms of aggregate metrics.

Table 6 (Appendix A.4) quantifies model robustness under spike and non-spike conditions. Deep learning models exhibit smaller relative degradation during spike periods; yet, this should be interpreted carefully as these methods incur relatively larger errors during non-spike conditions. Thus, smaller degradation does not necessarily imply stronger absolute performance under nonstationarity. The two best individual methods, *CING-LEAR* and *Chronos 2 w.*, show robustness under extreme price events. Their ensemble achieves the strongest performance under both regimes. These results suggest covariate support plays an important role in maintaining forecast stability under nonstationarity.

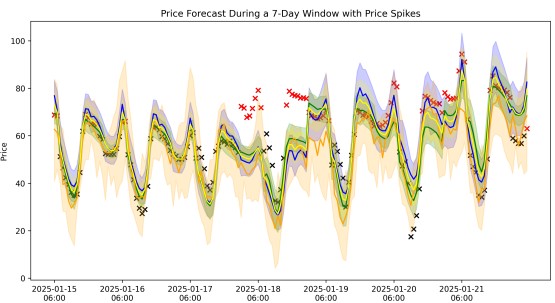

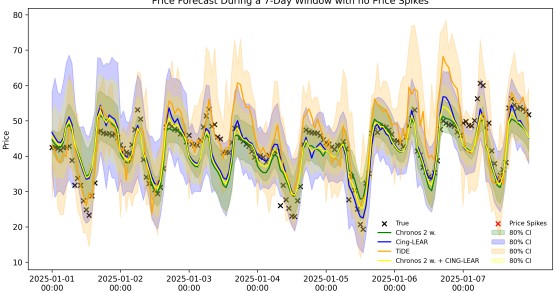

*Figure 1.* Performance of representative models from each model family for a 7-day period containing the most (top panel) and the least (bottom panel) price spikes on the GridStatus2025 dataset.

## 4. Conclusion

This study benchmarks TSFMs for electricity price forecasting (EPF) under contamination risk, distributional shifts, tail events, and strong domain dependence. Across two datasets, TSFMs show competitive zero-shot and probabilistic forecast performance, especially when supported by exogenous covariates. Our results suggest that accurate and robust performance in EPF is strongly tied to domain structure and task-specific inductive biases. Meanwhile, domain-specific methods remain highly competitive, especially under extreme price regimes and nonstationary market conditions.

This study highlights the importance of contamination-conscious and distribution-aware evaluation protocols, especially in structured domains such as EPF. Our results suggest that TSFMs are highly promising, and that domain-informed design is important in complex systems. Interestingly, ensembles of TSFMs and domain-aware models are highly competitive, suggesting that combinations of domain-informed models and TSFMs is a fruitful path to pursue.

These findings suggest several future directions: fine-tuning TSFMs for EPF-specific objectives, such as tail accuracy, is a promising step toward value-oriented forecasting. Another direction is to develop stronger ensembles that combine the broad generalization of TSFMs with domain-aware models tailored to energy applications.

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

# A. Appendix

## A.1. Comparative Listing of Prior Times Series Foundation Models Benchmarking Efforts

*Table 2.* Comparison of existing benchmarking efforts for TSFMs. $\checkmark$ indicates full inclusion, $\times$ represents no inclusion, and $\sim$ is partial inclusion. The date entry is based on their first draft submission date on Arxiv (or date provided by publisher where applicable).

| BENCHMARKING STUDIES | TSFMS INCLUSION | DEEP LEARNING BASELINES | EPF FOCUSED | COVARIATE ANALYSIS | DOMAIN-AWARE METHODS | SPIKE ANALYSIS | TAIL ANALYSIS | DATE |
|---|---|---|---|---|---|---|---|---|
| PROBTS (ZHANG ET AL., 2024) | $\checkmark$ | $\checkmark$ | $\times$ | $\times$ | $\times$ | $\times$ | $\times$ | 10/23 |
| TFB (QIU ET AL., 2024) | $\checkmark$ | $\checkmark$ | $\times$ | $\times$ | $\times$ | $\times$ | $\times$ | 03/24 |
| GIFT-EVAL (AKSU ET AL., 2024) | $\checkmark$ | $\checkmark$ | $\times$ | $\times$ | $\times$ | $\times$ | $\times$ | 10/24 |
| TSFM-BENCH (LI ET AL., 2025) | $\checkmark$ | $\checkmark$ | $\times$ | $\times$ | $\times$ | $\times$ | $\times$ | 10/24 |
| TSFMS FOR ELECTRICITY PRICE FORECASTING (HORNEK ET AL., 2025) | $\checkmark$ | $\times$ | $\checkmark$ | $\times$ | $\times$ | $\times$ | $\times$ | 07/25 |
| FEV-BENCH (SHCHUR ET AL., 2025) | $\checkmark$ | $\times$ | $\times$ | $\sim$ | $\times$ | $\times$ | $\times$ | 09/25 |
| TS-ARENA (MEYER ET AL., 2025) | $\checkmark$ | $\sim$ | $\sim$ | $\times$ | $\times$ | $\times$ | $\times$ | 02/26 |
| TIME (QIAO ET AL., 2026) | $\checkmark$ | $\times$ | $\times$ | $\times$ | $\times$ | $\times$ | $\times$ | 02/26 |
| TEMPUSBENCH (GOKTAS ET AL., 2026) | $\checkmark$ | $\checkmark$ | $\times$ | $\sim$ | $\times$ | $\times$ | $\times$ | 04/26 |
| **THIS STUDY** | $\checkmark$ | $\checkmark$ | $\checkmark$ | $\checkmark$ | $\checkmark$ | $\checkmark$ | $\checkmark$ | — |

## A.2. Evaluation Metrics

Let $t_r$ denote the forecast origin of rolling window $r$, where $r = 1, \ldots, \mathscr{R}$, and let $\delta_{t_r}$ denote the lead time between the forecast origin and the first hour of the forecast horizon. For each forecast origin, the model predicts the next $H$ prices over the target interval $t_r + \delta_{t_r} : t_r + \delta_{t_r} + H - 1$, and we write $y_{r,h} = Y_{t_r + \delta_{t_r} + h - 1}$ and $\hat{y}_{r,h} = \hat{Y}_{t_r + \delta_{t_r} + h - 1}$ for the observed and predicted prices at horizon $h = 1, \ldots, H$.

Mean Absolute Error (MAE) measures the average absolute deviation between the predicted and observed prices across all forecast horizons and rolling windows.

$$\text{MAE} = \frac{1}{\mathscr{R}} \sum_{r=1}^{\mathscr{R}} \left( \frac{1}{H} \sum_{h=1}^{H} |y_{r,h} - \hat{y}_{r,h}| \right) \tag{2}$$

Root Mean Square Error (RMSE) measures the square-rooted average squared prediction error, placing larger penalties on large forecasting mistakes and therefore emphasizing extreme errors.

$$\text{RMSE} = \sqrt{\frac{1}{\mathscr{R}} \sum_{r=1}^{\mathscr{R}} \frac{1}{H} \sum_{h=1}^{H} (y_{r,h} - \hat{y}_{r,h})^2} \tag{3}$$

Average Quantile Loss (aQL) evaluates probabilistic forecasting performance by averaging the quantile loss over a set of quantile levels and rolling windows. For notational simplicity, we define per-roll quantile loss in Equation 5, where $q$ denotes the evaluated quantile level and $r$ denotes the rolling window. Here, $\hat{y}_{r,h}^{(q)}$ and $y_{r,h}$ are the $q$-quantile forecast and actual observation for horizon $h$ in roll $r$, respectively. Quantile loss, also known as pinball loss, is defined in Equation 6.

$$\text{aQL}_{\mathscr{Q}} = \frac{1}{\mathscr{R}} \sum_{r=1}^{\mathscr{R}} \left( \frac{1}{|\mathscr{Q}|} \sum_{q \in \mathscr{Q}} \text{QL}_{q,r} \right) \tag{4}$$

$$\text{QL}_{q,r} = \frac{1}{H} \sum_{h=1}^{H} L_q \left( \hat{y}_{r,h}^{(q)}, y_{r,h} \right). \tag{5}$$

$$L_q(\hat{y}, y) = \begin{cases} (1-q)(\hat{y} - y), & y < \hat{y}, \\ q(y - \hat{y}), & y \geq \hat{y}. \end{cases} \tag{6}$$

## A.3. Implementation Details

On the GridStatus2025 dataset, we include a wide range of benchmark methods, including statistical models-*Auto-ARIMA*, *Auto-ETS*, and Auto-Theta (Herzen et al., 2022), and deep learning models—*DeepAR* (Salinas et al., 2020), *DLinear*, *NLinear* (Zeng et al., 2023), *TFT* (Lim et al., 2021), *TiDE* (Das et al., 2023a), where all methods are implemented with Darts (Herzen et al., 2022) and Optuna (Akiba et al., 2019).

After hyperparameter search using Optuna (Akiba et al., 2019) given the search space in Table 3, each deep learning method is trained up to 100 epochs with early stopping. Statistical models (except *Seasonal Naïve*) use future covariates, and deep learning models use both past and future covariates, with the exception that *DeepAR* only supports future covariates (Herzen et al., 2022). Unless otherwise specified, no transformations are applied to statistical or TSFMs inputs, while deep learning models use standard mean–variance normalization defined as $x' = \frac{x-\mu}{\sigma}$.

We additionally include domain-aware models: LEAR (Uniejewski et al., 2016), DNN (Lago et al., 2018), and CING-LEAR, which apply the transformation $\mathrm{asinh}(x) = \log(x + \sqrt{1 + x^2})$ after median-based scaling (Uniejewski et al., 2018; Ziel & Weron, 2018). This transformation is also applied to statistical models for consistency, while deep learning models use standard normalization and TSFMs rely on their internal preprocessing pipeline. Domain-aware models are trained on two years of data due to improved empirical performance, while all other general-purpose models use the full three years.

We fix the input context length of all TSFMs at $2048$ to ensure fair comparison. Prior work on Chronos 2 suggests that performance gains is marginal by extending the context length beyond 2048 (Ansari et al., 2025). Furthermore, the selected TSFMs differ in their supported maximum context lengths, thus, allowing each model to use its default context length would introduce an additional confounding factor. Therefore, we use a common context length of $2048$, which is reasonably long, so that all TSFMs are evaluated with the same amount of historical information.

*Table 3.* Search space for hyperparameters for deep learning models. For each model, we report the set of candidate values used in hyperparameter tuning, including input length, hidden size, number of layers, dropout rate, batch size, and learning rate where applicable. The optimal configuration is selected based on validation-set performance before final evaluation on the test set.

| | TSMIXER | | | | | | |
|---|---|---|---|---|---|---|---|
| **PARAMETERS** | INPUT_LENGTH | HIDDEN_SIZE | DROPOUT | BATCH_SIZE | LR | | |
| **SEARCH RANGE** | [168,336,720] | [64,128,256] | [0.0:0.3] | [16,32,64] | $[1e^{-5}:1e^{-1}]$ | | |
| | TIDE | | | | | | |
| **PARAMETERS** | INPUT_LENGTH | ENCODER_LAYERS | DECODER_LAYERS | DROPOUT | BATCH_SIZE | LR | |
| **SEARCH RANGE** | [168,336,720] | [1,2,3] | [1,2,3] | [0.0:0.3] | [16,32,64] | $[1e^{-5}:1e^{-1}]$ | |
| | TFT | | | | | | |
| **PARAMETERS** | INPUT_LENGTH | HIDDEN_SIZE | NUM_ATTENTION_HEAD | LSTM_LAYER | DROPOUT | BATCH_SIZE | LR |
| **SEARCH RANGE** | [168,336,720] | [64, 128, 256] | [2,4] | [1,2] | [0.0:0.3] | [16,32,64] | $[1e^{-5}:1e^{-1}]$ |
| | DEEPAR | | | | | | |
| **PARAMETERS** | INPUT_LENGTH | HIDDEN_SIZE | BATCH_SIZE | LR | | | |
| **SEARCH RANGE** | [168,336,720] | [64, 128, 256] | [16,32,64] | $[1e^{-5}:1e^{-1}]$ | | | |
| | DLINEAR AND NLINEAR | | | | | | |
| **PARAMETERS** | INPUT_LENGTH | BATCH_SIZE | LR | | | | |
| **SEARCH RANGE** | [168,336,720] | [16,32,64] | $[1e^{-5}:1e^{-1}]$ | | | | |

## A.4. Supplementary Result in Electricity Price Forecasting

*Table 4.* Leaderboard for GEFcom2014-P dataset. Results are normalized against the competition benchmark to compute rankings. The first 16 rows (above the horizontal line) represent the contestant entries, while the remaining rows (below the horizontal line) correspond to TSFMs and other baseline entries. Each column (1-12) represents the normalized performance for one evaluated task. The "Rating" column represents the weighted performance over the 12 evaluated tasks. The "Provisional Rank" column shows the final rank for the competition.

| Name | 1 | 2 | 3 | 4 | 5 | 6 | 7 | 8 | 9 | 10 | 11 | 12 | Rating | Provisional Rank |
|---|---|---|---|---|---|---|---|---|---|---|---|---|---|---|
| Arkadiy Strelnikov | 0.521 | 0.565 | 0.237 | 0.411 | 0.806 | 0.857 | 0.724 | 0.936 | 0.985 | 0.176 | 0.349 | 0.677 | 61.9% | 12 |
| Benchmark - Price | 0.000 | 0.000 | 0.000 | 0.000 | 0.000 | 0.000 | 0.000 | 0.000 | 0.000 | 0.000 | 0.000 | 0.000 | 0.0% | 32 |
| C3 Green Team | 0.539 | 0.589 | 0.779 | 0.581 | 0.821 | 0.861 | 0.757 | 0.958 | 0.971 | -0.079 | 0.514 | 0.706 | 65.0% | 6 |
| E.S. Mangalova | 0.489 | 0.000 | 0.846 | 0.420 | 0.712 | 0.851 | 0.669 | 0.978 | 0.935 | 0.025 | 0.331 | 0.671 | 59.3% | 19 |
| EPSteam | 0.000 | 0.000 | 0.000 | 0.715 | 0.230 | 0.373 | 0.065 | 0.921 | 0.971 | 0.371 | 0.490 | 0.537 | 49.2% | 25 |
| Florencio Gonzalez | 0.000 | 0.457 | 0.237 | 0.200 | 0.726 | 0.001 | 0.780 | 0.851 | 0.829 | 0.370 | 0.327 | 0.803 | 54.8% | 21 |
| GMD | 0.075 | 0.776 | 0.838 | 0.581 | 0.838 | 0.913 | 0.729 | 0.953 | 0.961 | 0.278 | 0.337 | 0.694 | 67.1% | 4 |
| Manuel Oviedo de la Fuente | 0.596 | 0.589 | -1.408 | 0.450 | 0.386 | 0.774 | 0.637 | 0.929 | 0.966 | -0.279 | 0.000 | 0.789 | 42.5% | 30 |
| NimNid | 0.144 | -0.329 | 0.515 | -1.002 | 0.386 | 0.820 | 0.817 | 0.942 | 0.963 | 0.100 | 0.274 | 0.624 | 47.8% | 26 |
| San/Saini | 0.387 | 0.755 | 0.853 | 0.562 | 0.749 | 0.813 | 0.833 | 0.909 | 0.916 | -0.533 | 0.429 | 0.249 | 50.1% | 24 |
| Team Poland | 0.510 | 0.772 | 0.791 | 0.768 | 0.803 | 0.905 | 0.857 | 0.967 | 0.971 | -0.422 | 0.661 | 0.863 | 67.7% | 3 |
| Tololo | 0.576 | 0.818 | 0.806 | 0.834 | 0.761 | 0.894 | 0.912 | 0.976 | 0.943 | -0.037 | 0.580 | 0.841 | 71.7% | 1 |
| Xiaorong (Iris) Sun | 0.638 | 0.800 | 0.826 | 0.750 | 0.877 | 0.898 | 0.801 | 0.927 | 0.979 | -0.754 | 0.631 | 0.798 | 61.9% | 12 |
| Yanghai Cong | 0.578 | 0.267 | 0.153 | 0.738 | 0.733 | 0.854 | 0.695 | 0.879 | 0.898 | 0.492 | 0.528 | 0.347 | 61.8% | 14 |
| dmlab | 0.429 | 0.758 | 0.779 | 0.787 | 0.632 | 0.829 | 0.773 | 0.974 | 0.965 | -0.301 | -0.071 | 0.543 | 51.5% | 23 |
| pat1 | 0.413 | 0.751 | 0.812 | 0.770 | 0.890 | 0.894 | 0.539 | 0.960 | 0.948 | -0.288 | 0.669 | 0.720 | 64.5% | 7 |
| Chronos 2 w. | 0.634 | 0.817 | 0.781 | 0.493 | 0.702 | 0.818 | 0.794 | 0.942 | 0.976 | -0.613 | 0.452 | 0.737 | 62.8% | 9 |
| Chronos 2 | 0.543 | 0.818 | 0.830 | 0.457 | 0.706 | 0.799 | 0.607 | 0.882 | 0.972 | -0.655 | 0.666 | 0.715 | 61.2% | 15 |
| TimesFM 2.5 | 0.610 | 0.734 | 0.796 | 0.159 | 0.573 | 0.508 | 0.743 | 0.900 | 0.971 | -0.637 | 0.622 | 0.830 | 56.7% | 20 |
| TimesFM 2.5 w. | 0.339 | 0.787 | 0.670 | 0.391 | 0.660 | 0.614 | 0.660 | 0.860 | 0.956 | -0.211 | 0.638 | 0.838 | 60.0% | 17 |
| TOTO 1.0 w. | 0.619 | 0.766 | 0.792 | 0.464 | 0.625 | 0.686 | 0.648 | 0.909 | 0.973 | -0.377 | 0.558 | 0.811 | 62.3% | 11 |
| TOTO 1.0 | 0.644 | 0.782 | 0.785 | 0.431 | 0.600 | 0.668 | 0.718 | 0.898 | 0.970 | -0.261 | 0.571 | 0.816 | 63.5% | 8 |
| TabPFN-TS w. | 0.403 | 0.741 | 0.766 | 0.503 | 0.775 | 0.861 | 0.789 | 0.963 | 0.960 | -0.479 | 0.566 | 0.693 | 62.8% | 9 |
| TabPFN-TS | -1.027 | -0.234 | 0.234 | -1.047 | -0.073 | -0.049 | 0.055 | 0.823 | 0.887 | -0.941 | 0.019 | 0.279 | -9.0% | 33 |
| Auto-ARIMA | 0.180 | 0.497 | 0.529 | 0.347 | 0.734 | 0.834 | -0.044 | 0.919 | 0.940 | -0.642 | 0.169 | 0.815 | 44.0% | 29 |
| Auto-Theta | 0.19 | 0.549 | 0.291 | 0.403 | 0.501 | 0.472 | 0.755 | 0.906 | 0.931 | -0.288 | 0.07 | 0.766 | 46.2% | 27 |
| Auto-ETS | 0.165 | 0.518 | 0.263 | 0.374 | 0.51 | 0.461 | 0.75 | 0.914 | 0.938 | -0.318 | 0.098 | 0.769 | 45.4% | 28 |
| TFT | 0.509 | 0.516 | 0.811 | 0.617 | 0.531 | 0.534 | 0.853 | 0.96 | 0.969 | -0.368 | 0.58 | 0.722 | 60.3% | 16 |
| TSMixer | 0.584 | 0.56 | 0.705 | 0.681 | 0.733 | 0.681 | 0.508 | 0.931 | 0.983 | 0.085 | 0.682 | 0.822 | 66.3% | 5 |
| DeepAR | 0.272 | 0.584 | 0.834 | 0.8 | 0.855 | 0.787 | 0.858 | 0.95 | 0.916 | -0.827 | 0.526 | 0.601 | 59.6% | 18 |
| TiDE | 0.602 | 0.822 | 0.801 | 0.633 | 0.605 | 0.663 | 0.848 | 0.924 | 0.951 | -0.066 | 0.619 | 0.774 | 68.1% | 2 |
| DLinear | -0.698 | -0.07 | -0.007 | 0.167 | 0.423 | 0.421 | 0.465 | 0.646 | 0.875 | -1.137 | -0.522 | 0.477 | 8.7% | 31 |
| NLinear | 0.355 | 0.686 | 0.769 | 0.397 | 0.453 | 0.526 | 0.56 | 0.904 | 0.968 | -0.498 | 0.451 | 0.809 | 53.2% | 22 |

*Table 5.* Tail performance evaluation on six tail quantiles {0.01, 0.025, 0.05, 0.95, 0.975, 0.99}. Evaluation is performed using pinball loss, defined in Appendix A.2, at the given quantile, as averaged across forecast horizons and rolling windows. In addition to the tail quantiles, we also include the $50th$ quantile for comparison. Best results are bolded and second-best results are underlined.

| CATEGORY | MODEL | Q(0.01) | Q(0.025) | Q(0.05) | Q(0.50) | Q(0.95) | Q(0.975) | Q(0.99) |
|---|---|---|---|---|---|---|---|---|
| STATISTICAL | AUTO-ARIMA | 22.9717 | 24.3620 | 26.1230 | 2.3198 | 2.7168 | 1.4257 | 0.5895 |
| | AUTO-THETA | 9.3775 | 11.1608 | 13.8413 | 3.9561 | 3.6761 | 2.0352 | 0.8757 |
| | AUTO-ETS | 9.9395 | 11.8953 | 14.6385 | 4.7535 | 3.1153 | 1.7102 | 0.7315 |
| DEEP LEARNING | TFT | 0.2262 | 0.5220 | 0.9442 | 2.8426 | 1.3577 | 0.8663 | 0.4973 |
| | TSMIXER | 0.3500 | 0.5127 | 0.7843 | 3.3129 | 1.0443 | 0.7985 | 0.6526 |
| | DLINEAR | 0.3731 | 0.8784 | 1.6930 | 4.8234 | 1.5223 | 0.8241 | 0.3856 |
| | NLINEAR | 0.6286 | 0.8749 | 1.2853 | 5.0720 | 1.4910 | 0.9572 | 0.6431 |
| | DEEPAR | 1.0434 | 1.3240 | 1.6922 | 3.7711 | 1.3410 | 0.9567 | 0.6860 |
| | TIDE | 0.2911 | 0.5069 | 0.8626 | 2.8422 | 0.8607 | 0.5381 | 0.3421 |
| TSFM | CHRONOS 2 w. | 0.1633 | – | 0.5833 | 2.0530 | 0.6333 | – | 0.2036 |
| | CHRONOS 2 | 0.1954 | – | 0.6705 | 2.3490 | 0.7594 | – | 0.2580 |
| | TIMESFM 2.5 w. | – | – | – | 2.4912 | – | – | – |
| | TIMESFM 2.5 | – | – | – | 2.4859 | – | – | – |
| | TOTO 1.0 w. | 4.3079 | 4.9084 | 5.4851 | 5.7711 | 1.3466 | 0.8586 | 0.5148 |
| | TOTO 1.0 | 4.2691 | 4.8514 | 5.3399 | 5.8980 | 1.3822 | 0.9085 | 0.5689 |
| | TABPFN-TS w. | 0.1784 | 0.3632 | 0.6137 | 2.2610 | 0.6973 | 0.4165 | 0.2047 |
| | TABPFN-TS | 0.2091 | 0.4244 | 0.7237 | 2.6600 | 0.7694 | 0.4603 | 0.2346 |
| DOMAIN-AWARE | DNN | 0.1998 | 0.4069 | 0.6814 | 2.4710 | 0.7214 | 0.4276 | 0.2107 |
| | LEAR | 0.2277 | 0.4481 | 0.7378 | 2.5095 | 0.7476 | 0.4459 | 0.2244 |
| | CING-LEAR | 0.1796 | **0.3619** | 0.6031 | 2.1010 | 0.6346 | **0.3833** | 0.1981 |
| ENSEMBLE | CHRONOS 2 w. + CING-LEAR | **0.1507** | – | **0.5373** | **1.9608** | **0.5875** | – | **0.1825** |

*Table 6.* Forecasting performance across spike and non-spike periods for the GridStatus2025 dataset, where spike is defined as observed prices fall outside 90% intervals taken with respect to the test set. $\Delta(\%)$ represents the percentage change in respective error measure with respect to non-spike periods, calculated as $\frac{error\_spike - error\_nonspike}{error\_nonspike}$. The "Rank Diff." columns indicate the change in ranking between non-spike and spike periods (positive = worse, negative = better).

| | | MAE | | | | RMSE | | | | AQL | | | |
| CATEGORY | MODEL | NON SPIKE | SPIKE | $\Delta(\%)$ | RANK DIFF. | NON SPIKE | SPIKE | $\Delta(\%)$ | RANK DIFF. | NON SPIKE | SPIKE | $\Delta(\%)$ | RANK DIFF. |
|---|---|---|---|---|---|---|---|---|---|---|---|---|---|
| STATSTICAL | AUTO-ARIMA | 4.177 | 8.782 | 110.2 | 0 | 5.744 | 11.436 | 99.1 | +3 | 1.774 | 3.641 | 105.2 | -1 |
| | AUTO-THETA | 7.125 | 14.992 | 110.4 | +4 | 9.584 | 18.688 | 95.0 | +4 | 3.523 | 6.655 | 88.9 | +3 |
| | AUTO-ETS | 8.805 | 15.812 | 79.6 | +2 | 12.418 | 19.313 | 55.5 | 0 | 3.730 | 6.558 | 75.8 | +1 |
| DEEP LEARNING | TFT | 7.712 | 8.965 | 16.2 | -8 | 10.559 | 11.378 | 7.8 | -9 | 3.053 | 3.524 | 15.4 | -9 |
| | TSMIXER | 6.456 | 8.130 | 25.9 | -9 | 8.295 | 10.294 | 24.1 | -9 | 2.474 | 3.193 | 29.1 | -9 |
| | DLINEAR | 9.536 | 10.440 | 9.5 | -4 | 12.417 | 13.211 | 6.4 | -3 | 4.028 | 4.345 | 7.9 | -2 |
| | NLINEAR | 10.017 | 11.461 | 14.4 | -4 | 13.202 | 14.941 | 13.2 | -4 | 4.058 | 4.710 | 16.1 | -3 |
| | DEEPAR | 7.285 | 9.842 | 35.1 | -2 | 9.763 | 12.376 | 26.8 | -3 | 2.985 | 4.033 | 35.1 | -1 |
| | TIDE | 5.359 | 8.908 | 66.2 | -5 | 6.934 | 11.219 | 61.8 | -7 | 2.216 | 3.623 | 63.5 | -4 |
| TSFM | CHRONOS 2 W. | 3.706 | 7.693 | 107.6 | +1 | 5.033 | 9.955 | 97.8 | 0 | 1.464 | 3.058 | 108.9 | +1 |
| | CHRONOS 2 | 4.165 | 9.495 | 128.0 | +6 | 5.880 | 12.359 | 110.2 | +5 | 1.651 | 3.799 | 130.1 | +6 |
| | TIMESFM 2.5 W. | 4.511 | 9.227 | 104.5 | 0 | 6.125 | 11.947 | 95.1 | 0 | 1.757 | 3.656 | 108.1 | +1 |
| | TIMESFM 2.5 | 4.334 | 10.709 | 147.1 | +9 | 6.046 | 13.946 | 130.7 | +7 | 1.709 | 4.293 | 151.2 | +9 |
| | TOTO W. | 9.650 | 31.630 | 227.8 | +1 | 12.692 | 36.138 | 184.7 | +1 | 4.152 | 14.760 | 255.5 | -1 |
| | TOTO | 9.372 | 31.517 | 236.3 | +2 | 12.327 | 35.692 | 189.5 | +3 | 4.045 | 14.769 | 265.1 | +2 |
| | TABPFN-TS W. | 4.070 | 8.576 | 110.7 | +1 | 5.731 | 11.277 | 96.8 | +2 | 1.604 | 3.436 | 114.2 | +1 |
| | TABPFN-TS | 4.773 | 10.244 | 114.6 | +2 | 6.776 | 13.103 | 93.4 | +2 | 1.865 | 4.050 | 117.2 | +2 |
| DOMAIN-AWARE | DNN | 4.465 | 9.219 | 106.5 | 0 | 6.022 | 11.770 | 95.5 | +1 | 1.754 | 3.577 | 103.9 | 0 |
| | LEAR | 4.437 | 10.255 | 131.1 | +6 | 5.965 | 13.519 | 126.6 | +8 | 1.755 | 4.068 | 131.8 | +6 |
| | CING-LEAR | 3.823 | 7.620 | 99.3 | -1 | 5.121 | 10.266 | 100.5 | 0 | 1.509 | 3.057 | 102.6 | -1 |
| ENSEMBLE | CHRONOS 2 W. + CING-LEAR | **3.555** | **7.220** | 103.1 | 0 | **4.803** | **9.570** | 99.3 | 0 | **1.396** | **2.851** | 104.2 | 0 |

## A.5. Diebold-Mariano Test for Covariate Usage

In order to determine whether covariate usage leads to statistically significant improvement towards performance of TSFMs, we perform a one-sided Diebold-Mariano (DM) tests (Diebold & Mariano, 1995) at significance level $\alpha = 0.05$. For each evaluated TSFM, and each evaluation metric, MAE, RMSE, aQL, we compare the covariate-incorporated variant of the TSFMs against their covariate-free variants The test is conducted using roll-wise losses in correspondance to evaluation setup.

Let $\ell_r^w$ denote the loss of the covariate-informed TSFM on rolling window $r$, and let $\ell_r^u$ denote the loss of the corresponding covariate-free TSFM on the same rolling window. We define the loss differential as

$$d_r = \ell_r^w - \ell_r^u, \tag{7}$$

where $d_r < 0$ indicates that the covariate-informed TSFM achieves lower loss than their covariate-free counterpart. The null hypothesis is that covariate usage does not improve forecasting performance, while the one-sided alternative is that covariate usage reduces forecast loss:

$$H_0 : \mathbb{E}[d_r] \geq 0, \qquad H_1 : \mathbb{E}[d_r] < 0. \tag{8}$$

*Table 7.* Diebold-Mariano (DM) test results comparing TSFM variants with and without covariates for EPF. Each metric reports the rejection decision at significance level $\alpha = 0.05$ and the corresponding DM statistic. A checkmark represents rejection of the null hypothesis.

| Model | MAE | | RMSE | | aQL | |
|---|---|---|---|---|---|---|
| | Rejection of $H_0$ | DM Statistic | Rejection of $H_0$ | DM Statistic | Rejection of $H_0$ | DM Statistic |
| Chronos 2 | ✓ | -6.34 | ✓ | -7.04 | ✓ | -6.37 |
| TimesFM 2.5 | × | 0.16 | × | -0.88 | × | -0.84 |
| TOTO 1.0 | × | 11.62 | × | 12.02 | × | 9.98 |
| TabPFN-TS | ✓ | -5.15 | ✓ | -6.74 | ✓ | -5.09 |

