# OpenReview forum: "Evaluating Time Series Foundation Models for Electricity Price Forecasting: Contamination Risk, Distributional Shifts, and Covariate Dependence"
_ICML.cc/2026/Workshop/FMSD — FMSD @ ICML 2026 Poster_

### Official Review · Reviewer_hXNx · 2026-05-17
**clear written paper and careful experiment design on benchmarking TSFMs for electricity price forecasting but with no methodological innovation and limited datasets setup.**

**Rating:** 4
**Confidence:** 4

**Review:**

Overall evaluation:

This paper is clear written, well-structured and demonstrates careful experimental design and broad model coverage on benchmarking time series foundation models (also other types of models) for electricity price forecasting. However, benchmarking TSFMs itself is already well explored. And this paper shows no methodological innovation and limited datasets setup for benchmarking.

Pros：
1) Well-motivated problem with domain awareness and careful experiment design.
2) Fair comparison across model families Includes: statistical, deep learning, TSFMs, and domain-specific models.

Cons:
1) Limited methodological novelty, focusing on empirical benchmarking.
2) Limited datasets (only 2) from publicly available sources where contamination mitigation is unclear.

---

### Official Review · Reviewer_mCuw · 2026-05-19
**Good evaluation for electricity price forecasting but lacks novelty or insight**

**Rating:** 6
**Confidence:** 3

**Review:**

Strengths
- Comprehensive benchmarking with inclusion of domain specific baselines.
- The paper examines tail quantiles, spike robustness and probabilistic calibrations which is important for energy price forecasting.
- The contamination aware two dataset framework is a good design choice which might be useful for other researchers.



Areas of improvement:
- PJM, CAISO and ERCOT have different price dynamics thus, performance in one market does not tell us anything about model performance in other markets.
- The contamination argument lacks evidence. The authors do not confirm whether GEFCom2014-P is TSFM's pre-training corpus.
-Deep Learning baselines underperform which might indicate they require further hyperparameter tuning.
-There is a lack of algorithmic or conceptual novelty.

Detailed Comments:
- Include a second market to make the benchmarking more credible.
- Statistical tests like Diebold-Marino are required to indicate if performance differences are statistically meaningful.
- Further analysis of the results focussing on the importance of covariates is required.

The evaluation has been done carefully and focuses on important metrics for Electricity price forecasting. However, the work lacks novelty and in-depth analysis of results. Thus it is just above the acceptance threshold.

---

### Official Review · Reviewer_z3so · 2026-05-20
**Evaluating TSFMs on Electricity Price Forecasting**

**Rating:** 6
**Confidence:** 4

**Review:**

# Summary

This paper evaluates 4 time series foundation models (Chronos-2, TimesFM 2.5, TabPFN-TS, and TOTO) on Electricity Price Forecasting (EPF). Their results indicate that while TSFM can show strong forecasting performance they do not outperform SOTA domain-specific models. Additionally, they found covariate information to be crucial for effective forecasting.

# Strengths

* They perform an extensive evaluation measuring both point forecasts and probabilistic forecasts.
* They include additional analysis on both tail performance and performance under distributional shifts and price spikes.
* Experiments answer the research question of whether TSFM are good zero-shot forecasters on electricity price data. These results may be of interest to workshop attendees.

# Areas for Improvement \+ Detailed Comments

* Saying how model selection was guided is a bit vague. For example, your fourth point about potential exposure to energy-related data during pretraining, was the goal to avoid TSFM with energy in the pretraining data or was the goal to include them. Also, almost all SOTA TSFMs trained on real data include energy-related data and produce quantile forecasts.
* Section 2.3 starts off with the explanation of data contamination being a large problem, but for the models selected, there is documentation that they did not train on the evaluated datasets, so the emphasis on data contamination is unnecessary.
  * Chronos-2 and TimesFM lists all real training data, TOTO uses proprietary observability data and gift-eval/chronos datasets, and TabPFN-TS is synthetic only, none of which contain either EPF datasets
* The start of Section 3.1 states that TSFM are “strong” zero-shot forecasters on GEFCom2014-P, but this is followed up by saying none of them ranked highly and they were “insufficient in EPF.”
* The last paragraph of Section 3.1 questions if TOTO’s strong GEFCom2014-P performance but poor GridStatus2025 can be attributed to pretraining contamination which as described above is not a direct issue.
* As stated in the future research paragraph, I agree that the paper would benefit from an analysis of the performance benefits from fine-tuning as it is stated in Section 3.2 that feature engineering is highly effective on this task.